

# The Fingerprint of Climate Variability on the Surface Ocean Cycling of Iron and its Isotopes

Daniela König[1], Alessandro Tagliabue[1]

[1]School of Environmental Sciences, University of Liverpool, Liverpool, L69 3GP, UK

*Correspondence to*: D. König (koenigd@liverpool.ac.uk)

**Abstract.** The essential micronutrient iron (Fe) limits phytoplankton growth when dissolved Fe (dFe) concentrations are too low to meet biological demands. However, many of the processes that remove, supply, or transform Fe are poorly constrained, which limits our ability to predict how ocean productivity responds to ongoing and future changes in climate. In recent years, isotopic signatures ($\delta^{56}$Fe) of Fe have increasingly been used to gain insight into the ocean Fe cycle, as distinct $\delta^{56}$Fe

endmembers of external Fe sources and $\delta^{56}$Fe fractionation during processes such as Fe uptake by phytoplankton can leave a characteristic imprint on dFe signatures ($\delta^{56}$Fe$_{diss}$). However, given the relative novelty of these measurements, the temporal scale of $\delta^{56}$Fe$_{diss}$ observations is limited. Thus, it is unclear how the changes in ocean physics and biogeochemistry associated with ongoing or future climate change will affect $\delta^{56}$Fe$_{diss}$ on interannual to decadal time scales. To explore the response of $\delta^{56}$Fe$_{diss}$ to such climate variability, we conducted a suite of experiments with a global ocean model with active $\delta^{56}$Fe cycling

under two climate scenarios. The first scenario is based on an atmospheric reanalysis and includes recent climate variability (1958-2021), whereas the second comes from a historical and high emissions climate change simulation to 2100. We find that under recent climatic conditions (1975-2021), interannual $\delta^{56}$Fe$_{diss}$ variability is highest in the tropical Pacific due to circulation and productivity changes related to the El Niño Southern Oscillation (ENSO), which alter both endmember and uptake fractionation effects on $\delta^{56}$Fe$_{diss}$ by redistributing dFe from different external sources and shifting nutrient limitation patterns.

While the tropical Pacific remains a hotspot of $\delta^{56}$Fe$_{diss}$ variability in the future, the most substantial end of century $\delta^{56}$Fe$_{diss}$ changes occur in the Southern hemisphere at mid to high latitudes. These arise from uptake fractionation effects due to shifts in nutrient limitation. Based on these strong responses to climate variability, ongoing measurements of $\delta^{56}$Fe$_{diss}$ may help diagnose changes in external Fe supply and ocean nutrient limitation.

## 1 Introduction

The micronutrient iron (Fe) is thought to control primary productivity in large parts of the global ocean where limited supply and/or rapid removal keep dissolved Fe (dFe) concentrations low. However, our understanding of the ocean Fe cycle is still limited, as it involves a multitude of internal cycling processes and various supply mechanisms, both of which are often poorly constrained (Boyd and Ellwood, 2010; Tagliabue et al., 2017). One way to help disentangle the web of processes and sources is to measure the isotopic signatures of Fe (namely $\delta^{56}Fe[‰] = \left[\left(^{56}Fe/^{54}Fe\right)_{sample} / \left(^{56}Fe/^{54}Fe\right)_{standard} - 1\right] * 1000$),





as some Fe cycle processes distinctively alter $\delta^{56}$Fe via fractionation and many external sources of Fe have characteristic $\delta^{56}$Fe endmember signatures. Thus, $\delta^{56}$Fe have been used to study various aspects of the ocean cycle. For instance, changes in $\delta^{56}$Fe of the dissolved and particulate Fe pools have been observed for different phytoplankton bloom stages, indicating that phytoplankton may preferentially take up light Fe, so that the $\delta^{56}$Fe of the dFe pool ($\delta^{56}$Fe$_{diss}$) becomes increasingly heavy (Ellwood et al., 2015). Consequently, heavy $\delta^{56}$Fe$_{diss}$ observed in other low dFe surface ocean systems were suggested to be

due to on-going phytoplankton uptake and biological recycling, possibly combined with fractionation during the complexation of Fe by organic ligands (Ellwood et al., 2020; Sieber et al., 2021). On the other hand, the distinct $\delta^{56}$Fe endmembers of external sources have been used in mass balance approaches (of $\delta^{56}$Fe$_{diss}$ and dFe) to estimate the relative importance of each source in supplying new Fe (Conway and John, 2014; Pinedo-González et al., 2020). Such approaches take advantage of the wide range of source $\delta^{56}$Fe endmembers from exceedingly light $\delta^{56}$Fe associated with reductive sediments (as low as -3.3‰;

Homoky et al., 2009; Severmann et al., 2010) and aeolian Fe from anthropogenic emissions (as low as -4.0 ‰, Kurisu et al., 2019) to crustal $\delta^{56}$Fe (ca. 0.1‰) for dFe supplied by non-reductive sedimentary processes or dust deposition (Conway et al., 2019; Homoky et al., 2013; Radic et al., 2011; Waeles et al., 2007).

Overall, $\delta^{56}$Fe$_{diss}$ is likely determined by a combination of $\delta^{56}$Fe fractionation during internal cycling and supply of new Fe from external sources with characteristic $\delta^{56}$Fe endmembers (König et al., 2021, 2022). Both variable source endmembers and

fractionation during phytoplankton uptake and organic complexation are needed for a global Fe isotope model to reasonably reproduce $\delta^{56}$Fe$_{diss}$ observations (König et al., 2021). However, most $\delta^{56}$Fe$_{diss}$ observations and the majority of dFe concentration measurements have been obtained from single occupations of stations as part of GEOTRACES transects, and from a few process studies, one of which focuses on seasonal changes (Ellwood et al., 2015). Consequently, there is only limited observation-based information available about the temporal variability in $\delta^{56}$Fe$_{diss}$ (and Fe cycling in general), and their

response to changes in ocean physics driven by climate variability. So far, only one study has reported repeat $\delta^{56}$Fe$_{diss}$ measurements from re-occupations of three different stations in the Atlantic (Conway et al., 2016). While at each station the shape of the $\delta^{56}$Fe$_{diss}$ profiles in this work was similar for both occupations, some discrepancies were detected for the station near Cape Verde and the upper part of the profile in the Cape Basin. Variable $\delta^{56}$Fe$_{diss}$ near Cape Verde were suggested to relate to changes in the relative contribution of dFe from different external sources, namely Fe supply by reductive sediments

and dust dissolution. In the Cape Basin, changes in surface currents were thought responsible, as the differences in upper ocean $\delta^{56}$Fe$_{diss}$ and dFe coincided with changes in temperature, oxygen, and salinity. These observations suggests that $\delta^{56}$Fe$_{diss}$ can vary on interannual time scales, especially in the upper ocean. Substantial temporal variability in local Fe sources and cycling has also been observed for repeat measurements of Fe concentration at time series stations in the North Pacific, Atlantic, and Mediterranean (Bonnet and Guieu, 2006; Fitzsimmons et al., 2015; Nishioka et al., 2001; Schallenberg et al., 2015; Sedwick

et al., 2005, 2020), but no parallel Fe isotope measurements were made.

Modelling work suggests that seasonal variability in dFe supply from aeolian deposition and winter mixing can induce variability in surface ocean $\delta^{56}$Fe$_{diss}$ both directly and indirectly (König et al., 2022). Direct effects occur due to the variable $\delta^{56}$Fe endmembers of each source, and indirect effects also arise as differences in Fe supply can change the degree of $\delta^{56}$Fe



fractionation during phytoplankton uptake and/or complexation by organic ligands. The change in the degree of uptake
fractionation is suggested to be highest in areas where Fe was the limiting nutrient of primary production. However, while
aeolian Fe deposition varied between years in this study, changes in ocean physics driven by climate variability were not
accounted for, as the same (seasonally variable) physical forcing fields were applied each year. We would expect climate-
driven changes in ocean circulation to induce similar, if not larger, changes in $\delta^{56}Fe_{diss}$ than variable external Fe supply, as
such changes can redistribute dFe and other nutrients more significantly. Changes in nutrient availability and other parameters
(e.g., temperature) can then lead to shifts in local upper ocean biogeochemistry, potentially altering the effect of fractionating
processes on $\delta^{56}Fe_{diss}$, for instance during phytoplankton uptake (Ellwood et al., 2015). On interannual time scales, climate-
related changes in ocean physics are mainly driven by climate variations such as the Southern Annular Mode or the El Niño-
Southern Oscillation (ENSO), which has been shown to impact Fe cycling in the Californian Current System (Johnson et al.,
1999). On longer times scales, global warming will likely lead to changes in ocean physics and, consequently, biogeochemistry,
whereby the extent of such effects depends on current and future $CO_2$ emissions (Bindoff et al., 2019; Kwiatkowski et al.,
2020), but their effects on Fe and Fe isotopes remain poorly constrained.

To test how surface ocean $\delta^{56}Fe_{diss}$ responds to interannual and decadal climate variability, we set up two sets of model
experiments. In a set of "hindcast" simulations, variability in ocean physics was driven by the inherent variability of an
atmospheric reanalysis product, covering the years 1958 to 2021. In a second set of experiments, we used outputs from a 249-
year climate change simulation (1852-2100), which includes internal climate variability as well as effects of long term global
warming. These experiments allowed us to detect hotspots of surface ocean $\delta^{56}Fe_{diss}$ variability in the present and future climate
and identify the driving forces behind such variability.

## 2 Methods

### 2.1 Experimental design

We conducted both sets of experiments (referred to as "hindcast" and "climate change") by applying off-line physical forcing
fields to a version of the PISCES biogeochemical ocean model with active $\delta^{56}Fe$ cycling, which has been used previously to
study the effect of internal cycling and (variable) external $\delta^{56}Fe$ endmembers on $\delta^{56}Fe_{diss}$ (König et al., 2021, 2022). This model
is based on PISCES-v2 (Aumont et al., 2015) which represents two phytoplankton, two zooplankton, and two particle size
classes (with variable particle reactivity; Aumont et al., 2017), five nutrients ($NO_3$, $NH_4$, $PO_4$, Fe, and Si), oxygen, and
carbonate system. Its Fe cycle representation is comparably complex (Tagliabue et al., 2016), and includes a prognostic Fe-
binding ligand tracer (Völker and Tagliabue, 2015) and Fe input from four external sources (rivers, hydrothermal vents, dust
deposition, sediments) plus sea ice (source or sink; Aumont et al., 2015). In the $\delta^{56}Fe$ version of this model, each Fe tracer is
split into a heavy ($^{56}Fe$) and light ($^{54}Fe$) pool, and distinct $\delta^{56}Fe$ endmembers are applied to each of the four external Fe sources
(König et al., 2021). In addition, the model applies $\delta^{56}Fe$ fractionation to two of the internal cycling processes: complexation
by organic ligands (preference for heavy $^{56}Fe$) and uptake by phytoplankton (preference for light $^{54}Fe$; König et al., 2021).



For the hindcast model experiments, we applied 5-daily physical forcing fields obtained from coupled NEMO-SI$^3$-PISCES simulations forced by the JRA-55 atmospheric reanalysis (Tsujino et al., 2018), as described in Buchanan and Tagliabue (2021), extended to 2021. We conducted three repeat cycles of the 64-year period (1830-2021). As changes in climate in the latter years of the cycle may impact the earlier years of the last repeat cycle, we focus on the last 46 years of each simulation
100 (1975-2021).

To investigate the impact of climate change on $\delta^{56}Fe_{diss}$, we set up climate change simulations using offline forcing fields from the IPSL-CM5A climate model (Dufresne et al., 2013), as described in previous studies (e.g., Richon and Tagliabue, 2021). For each experiment (i.e., standard simulation and sensitivity tests), we first set up pre-industrial (PI) control simulations from 1801 to 2100 using forcings from an IPSL-CM5A experiment with fixed pre-industrial atmospheric $CO_2$. We then set up
climate change experiments from 1852 to 2100, initialised using the 1851 PI control output and forced by fields from IPSL-CM5A experiments with atmospheric $CO_2$ concentrations following historical pathways until 2005, and switching to the high emissions RCP8.5 scenario thereafter.

## 2.2 Sensitivity experiments and isolating $\delta^{56}Fe_{diss}$ components

We conducted an equivalent set of "standard" $\delta^{56}Fe$ model simulations and sensitivity tests for both hindcast and climate
change physical settings (Table 1). For the "standard" simulations, we used the $\delta^{56}Fe$ model set-up described in König et al. (2021), whereas for the sensitivity experiments we turned off either uptake or complexation fractionation by setting their respective fractionation factors to 1. This allowed us to estimate the extent of $\delta^{56}Fe_{diss}$ variability that is caused by changes in fractionation and source endmember effects, using the following set of calculations.

**Table 1. Overview of experiments**

| Name | Physical forcings | Duration | Analysed | $\delta^{56}Fe$ endmembers | $\delta^{56}Fe$ fractionation factors |
|---|---|---|---|---|---|
| *Hindcast standard* | JRA-55 | 1830-2021 | 1975-2021 | *Standard*: Dust: +0.09‰, Rivers: 0‰, Hydrothermal vents: -0.5‰, Sediments: -1‰ to ±0.09‰ | *Standard*: Phytoplankton uptake: 0.9995, Organic complexation: 1.0006 |
| *Hindcast noUF* | JRA-55 | 1830-2021 | 1975-2021 | Standard | Phytoplankton uptake: 1.0 |
| *Hindcast noCF* | JRA-55 | 1830-2021 | 1975-2021 | Standard | Organic complexation: 1.0 |
| *Hindcast neuSED* | JRA-55 | 1830-2021 | 1975-2021 | Sediments: -1‰ | Standard |
| *PI control standard* | IPSL-CM5A (PI control) | 1801-2100 | 2006-2100 | Standard | Standard |
| *Climate change standard* | IPSL-CM5A (historical + RCP8.5) | 1852-2100 | 2006-2100 | Standard | Standard |
| *Climate change noUF* | IPSL-CM5A (historical + RCP8.5) | 1852-2100 | 2006-2100 | Standard | Phytoplankton uptake: 1.0 |
| *Climate change noCF* | IPSL-CM5A (historical + RCP8.5) | 1852-2100 | 2006-2100 | Standard | Organic complexation: 1.0 |





Firstly, we "split up" $\delta^{56}Fe_{diss}$ into uptake fractionation ($\delta^{56}Fe_{UF}$), complexation fractionation ($\delta^{56}Fe_{CF}$), and endmember ($\delta^{56}Fe_{EM}$) contributions, i.e., the effect which each fractionation process and the source endmembers have on "overall" $\delta^{56}Fe_{diss}$. In the case of $\delta^{56}Fe_{UF}$ and $\delta^{56}Fe_{CF}$, we calculated their respective contributions using the $\delta^{56}Fe_{diss}$ of the standard experiment, and those obtained from sensitivity experiments where either the uptake fractionation factor ("noUF") or complexation fractionation factor ("noCF") was set to 1 (Eq. 1 and 2).

$\delta^{56}Fe_{UF} = \delta^{56}Fe_{diss} - \delta^{56}Fe_{diss,\ noUF}$                                                 (1)

  $\delta^{56}Fe_{CF} = \delta^{56}Fe_{diss} - \delta^{56}Fe_{diss,\ noCF}$                                                 (2)

This then allowed us to calculate the endmember effect, i.e., $\delta^{56}Fe_{EM}$, by subtracting the two fractionation effects (Eq. 3).

  $\delta^{56}Fe_{EM} = \delta^{56}Fe_{diss} - \delta^{56}Fe_{UF} - \delta^{56}Fe_{CF}$                                       (3)

Note that for simplicity, we did not conduct a sensitivity experiment with endmember effects turned off (i.e., all $\delta^{56}Fe$
endmembers set to 0‰) as in previous studies (König et al., 2021, 2022), which would allow calculating $\delta^{56}Fe_{EM}$ in a similar way as for $\delta^{56}Fe_{UF}$ and $\delta^{56}Fe_{CF}$. We did, however, set up an experiment ("neuSED") where only the sediment $\delta^{56}Fe$ endmember was set to 0‰ (only for the hindcast physical setting; Table 1) to study endmember effects in areas where the surface ocean sediment endmember of the model (-1‰; König et al., 2021) is likely too light (Sect. 4.1).

### 2.3 Determining interannual variability in $\delta^{56}Fe_{diss}$ and its components

To quantify the interannual variability in $\delta^{56}Fe_{diss}$ and its three components ($\delta^{56}Fe_{UF}$, $\delta^{56}Fe_{UF,}$, and $\delta^{56}Fe$), we first applied a 12-month running mean boxcar filter to each value (which were based on monthly model output) to smooth out seasonal effects. We then calculated the interannual standard deviation (SD) for $\delta^{56}Fe_{diss}$ and for each component over the respective time periods of interest (1975 to 2021 for the hindcast and 2006-2100 for the climate change experiments). Note that we focus here on the surface ocean (0-10m), as $\delta^{56}Fe_{diss}$ variability beneath the surface is often driven by small lateral or vertical movement
in steep $\delta^{56}Fe_{diss}$ gradients, for instance, due to shifts in mixed layer depth. To estimate the contribution of each of the three components to $\delta^{56}Fe_{diss}$ variability, we calculated the ratio between their SD and the sum of all three SD, where subscript i=UF, CF or EM (Eq. 4).

$$\delta^{56}Fe_i\ contribution = \frac{SD\ \delta^{56}Fe_i}{SD\ \delta^{56}Fe_{UF} + SD\ \delta^{56}Fe_{CF} + SD\ \delta^{56}Fe_{EM}}$$                       (4)





## 3 Results & Discussion

### 3.1 Hotspots of present and future $\delta^{56}Fe_{diss}$ variability in the surface ocean

#### 3.1.1 Areas of high $\delta^{56}Fe_{diss}$ variability in the present climate (1975-2021)

Figure 1: Surface ocean $\delta^{56}Fe_{diss}$ interannual variability and its drivers in the present climate (1975-2021). Surface ocean (0-10m) (a) interannual $\delta^{56}Fe_{diss}$ SD (‰) and (b) average $\delta^{56}Fe_{diss}$ (‰) of the hindcast standard experiment. Respective contributions of (c) $\delta^{56}Fe_{UF}$, (d) $\delta^{56}Fe_{EM}$, and (e) $\delta^{56}Fe_{CF}$ to $\delta^{56}Fe_{diss}$ SD. (f) Map of which driver(s) is locally dominant (i.e., contributing over 40% of the sum of SD); drivers are denoted as 'endmembers' (EM), 'uptake fractionation' (UF), 'complexation fractionation' (CF) and various combinations thereafter.



In the 1975-2021 period of the hindcast simulations, interannual surface ocean $\delta^{56}Fe_{diss}$ variability is highest in the tropical
Pacific due to multiple drivers. This is illustrated by maxima in the interannual $\delta^{56}Fe_{diss}$ SD, with additional areas of localised
elevated variability present around 40°S (Fig. 1a). Comparing this $\delta^{56}Fe_{diss}$ SD to the average $\delta^{56}Fe_{diss}$ values (Fig. 1b) indicates
that the $\delta^{56}Fe_{diss}$ SD is often elevated in areas with steep horizontal $\delta^{56}Fe_{diss}$ gradients, with shifts in circulation causing local
changes in $\delta^{56}Fe_{diss}$. The high $\delta^{56}Fe_{diss}$ variability in the tropical Pacific during our simulations is due to changes in both uptake
fractionation and the circulation of external source $\delta^{56}Fe$ endmembers, which are generally the dominant causes of $\delta^{56}Fe_{diss}$
variability at low and mid latitudes (Fig. 1c-f, Fig. S1). At high latitudes, a combination of changes in uptake and, to a lesser
extent, complexation fractionation dominates, although $\delta^{56}Fe_{diss}$ variability is generally lower in these areas (Fig. 1a). The
elevated $\delta^{56}Fe_{diss}$ SD at around 40°S is caused by horizontal shifts of areas with heavy $\delta^{56}Fe_{diss}$ caused by strong uptake
fractionation effects (Fig. 1c, Fig. S1d). Note that for some regions, including areas of high $\delta^{56}Fe_{diss}$ variability, temporal
variability in $\delta^{56}Fe_{UF}$, $\delta^{56}Fe_{CF}$, and/or $\delta^{56}Fe_{EM}$ (partially) cancel each other out, so that the overall $\delta^{56}Fe_{diss}$ SD is smaller than
the sum of $\delta^{56}Fe_{UF}$, $\delta^{56}Fe_{CF}$, and $\delta^{56}Fe_{EM}$ SD (Fig. S2). Furthermore, $\delta^{56}Fe_{diss}$ variability (unsurprisingly) increases when
seasonal effects are accounted for (by calculating $\delta^{56}Fe_{diss}$ SD from monthly model outputs without applying a 12-month
running mean; Fig. S3). Seasonal effects are thereby largest at high latitudes mostly due to uptake fractionation that now
dominate $\delta^{56}Fe_{diss}$ SD, especially in the southern hemisphere (south of ca. °40 S). However, a combination of endmember
changes due to ocean circulation and uptake fractionation remain important in the tropical Pacific (Fig. S3).

### 165 3.1.2 Future $\delta^{56}Fe_{diss}$ variability (2006-2100) due to climate change

The major changes in ocean circulation and physical conditions in our high emissions climate change experiments lead to
substantial $\delta^{56}Fe_{diss}$ variability over the next century. The interannual $\delta^{56}Fe_{diss}$ SD over the 2006-2100 period of our climate
change simulation is considerably higher than for the parallel PI control simulation without any climate change effects (and
the hindcast simulations, Fig. 2a,b). The $\delta^{56}Fe_{diss}$ SD of the PI control thereby resembles that of the standard hindcast
experiment (Fig. 1a; note difference in scale), with some discrepancies due to the differences in model circulation.

In the tropical Pacific, where interannual $\delta^{56}Fe_{diss}$ SD was highest over the period covered by the hindcast experiment (1975-
2021; Fig. 1a), variability remains high under future climate change conditions (Fig. 2a). However, while the magnitude of
$\delta^{56}Fe_{diss}$ SD is similar for the climate change experiment, there are some regional differences. $\delta^{56}Fe_{diss}$ SD is elevated in the
eastern equatorial Pacific, compared to the PI control, but lower in the western part and subtropical regions (Fig. 2c). Generally,
a combination of both endmember ($\delta^{56}Fe_{EM}$) and uptake fractionation ($\delta^{56}Fe_{UF}$) effects is responsible for the $\delta^{56}Fe_{diss}$ variability
in the tropical Pacific (Fig. 2d, Fig. S4), similar to the hindcast experiment (Fig. 1f), although in the eastern equatorial Pacific
where $\delta^{56}Fe_{diss}$ variability is increased, $\delta^{56}Fe_{EM}$ appears to be the controlling factor (see also Sect. 3.2.3).

A notable increase in interannual $\delta^{56}Fe_{diss}$ variability due to climate change occurs in the southern hemisphere. In the region
between ca. 30 to 50°S (Fig. 2a,b), where $\delta^{56}Fe_{diss}$ SD was already elevated under present climate conditions (hindcast
simulation; Fig. 1a), $\delta^{56}Fe_{diss}$ SD more than doubles due to climate change (Fig. 2c). While some of this variability is due to
endmember effects ($\delta^{56}Fe_{EM}$), changes in uptake fractionation ($\delta^{56}Fe_{UF}$) dominate the future $\delta^{56}Fe_{diss}$ variability in most of the





southern hemisphere at mid to high latitudes (Fig. 2d, Fig. S4). This is due to changes in $\delta^{56}Fe_{UF}$ between 2006 and 2100 caused by the impact of climate change on primary production (Sect. 3.3).

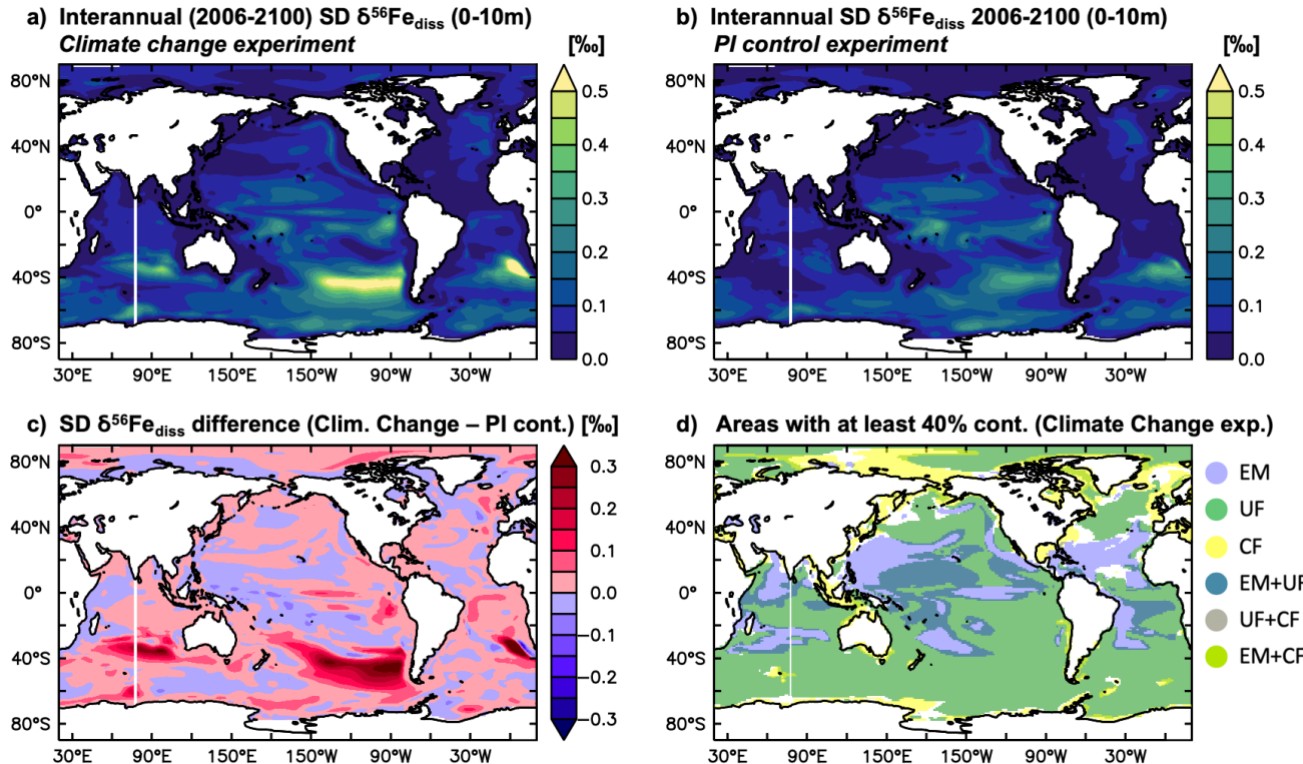

**Figure 2: Surface ocean $\delta^{56}Fe_{diss}$ interannual variability under future climate scenarios (2006-2100). Surface ocean (0-10m) interannual SD $\delta^{56}Fe_{diss}$ (‰) for (a) climate change and (b) PI control experiments, and (c) the difference between the two (i.e., climate change SD $\delta^{56}Fe_{diss}$ - PI control SD $\delta^{56}Fe_{diss}$; ‰). (d) Map of which driver(s) is locally dominant (i.e., contributing over 40% of the sum of SD $\delta^{56}Fe_{diss}$) for the climate change experiment.**

## 3.2 $\delta^{56}Fe_{diss}$ variability in the tropical Pacific

### 3.2.1 Current (1975-2021) $\delta^{56}Fe_{diss}$ variability in the tropical Pacific driven by ENSO variability

In the tropical Pacific, where $\delta^{56}Fe_{diss}$ variability in the hindcast simulation is highest (Fig. 1) there is a clear link between surface ocean $\delta^{56}Fe_{diss}$ and changes in sea surface temperature (SST) associated with different ENSO phases. This can be illustrated by focussing on the equatorial region between 5°S and 5°N (Fig. 3). As the hindcast simulations were forced by an atmospheric reanalysis product (see Methods), the modelled changes in SST agree very well with observations in the equatorial Pacific, including the timing and extent of the El Niño and La Niña transitions (Fig. S5).

In neutral or weak El Niño/La Niña phases, when SST anomalies and Ocean Nino Index (ONI) are less than ca. ± 0.5°C (Fig. 3a), $\delta^{56}Fe_{diss}$ is light (<-0.2‰) close to the continental margins of Papua New Guinea (PNG) in the west and South America in the east, and heavier in the offshore regions of the open ocean (up to >1‰; Fig. 3c). However, during an El Niño event, the



area of light surface ocean $\delta^{56}\text{Fe}_\text{diss}$ close to the PNG margin expands eastward (Fig. 3c), with a similar timing and extent as

the characteristic warming signal observed in the central or eastern equatorial Pacific (Fig. 3a,b). While these changes are

visible in all El Niño years, they are most prominent for the very strong El Niño events of 1982/83, 1997/98, and 2015/2016.

During these three extreme events (dark red ONI), surface ocean $\delta^{56}\text{Fe}_\text{diss}$ becomes anomalously light in most areas along the

equator, except in the easternmost part (around 120°W) where it is heavier (Fig. 3c). For weaker El Niños (light red ONI) and

those where the maximum SST anomalies occur in the centre of the equatorial Pacific, the light $\delta^{56}\text{Fe}_\text{diss}$ anomaly is restricted

to the western part of the equatorial Pacific (Fig. 3c). Conversely, La Niñas events (grey ONI) are associated with a $\delta^{56}\text{Fe}_\text{diss}$

decrease in the eastern part of the basin, albeit to a lesser degree than for strong El Niños, as the light $\delta^{56}\text{Fe}_\text{diss}$ area close to the

South American margin expands westward (Fig. 3c).

**Figure 3: Variability in SST, $\delta^{56}\text{Fe}_\text{diss}$ and its drivers in the equatorial Pacific from the hindcast experiment. Time series (1975-2021)**
**of monthly-mean surface ocean (a) SST anomaly (°C) and Ocean Nino Index (ONI; red: El Niño, Grey: La Niña), (b) SST (°C), (c)**
**$\delta^{56}\text{Fe}_\text{diss}$ (‰), (d) $\delta^{56}\text{Fe}_\text{EM}$ (‰), and (e) $\delta^{56}\text{Fe}_\text{UF}$ (‰), averaged from 5°N to 5°S. The SST anomaly was calculated by subtracting the**
**1975-2021 average SST from the monthly outputs. The Ocean Nino Index uses the same key as the SST anomaly and was calculated**
**from smoothed SST anomalies (3 month running mean) of the ENSO 3.4 region (120°-170°W, 5°N-5°S).**





### 3.2.2 Mechanisms behind ENSO-driven $\delta^{56}Fe_{diss}$ variability

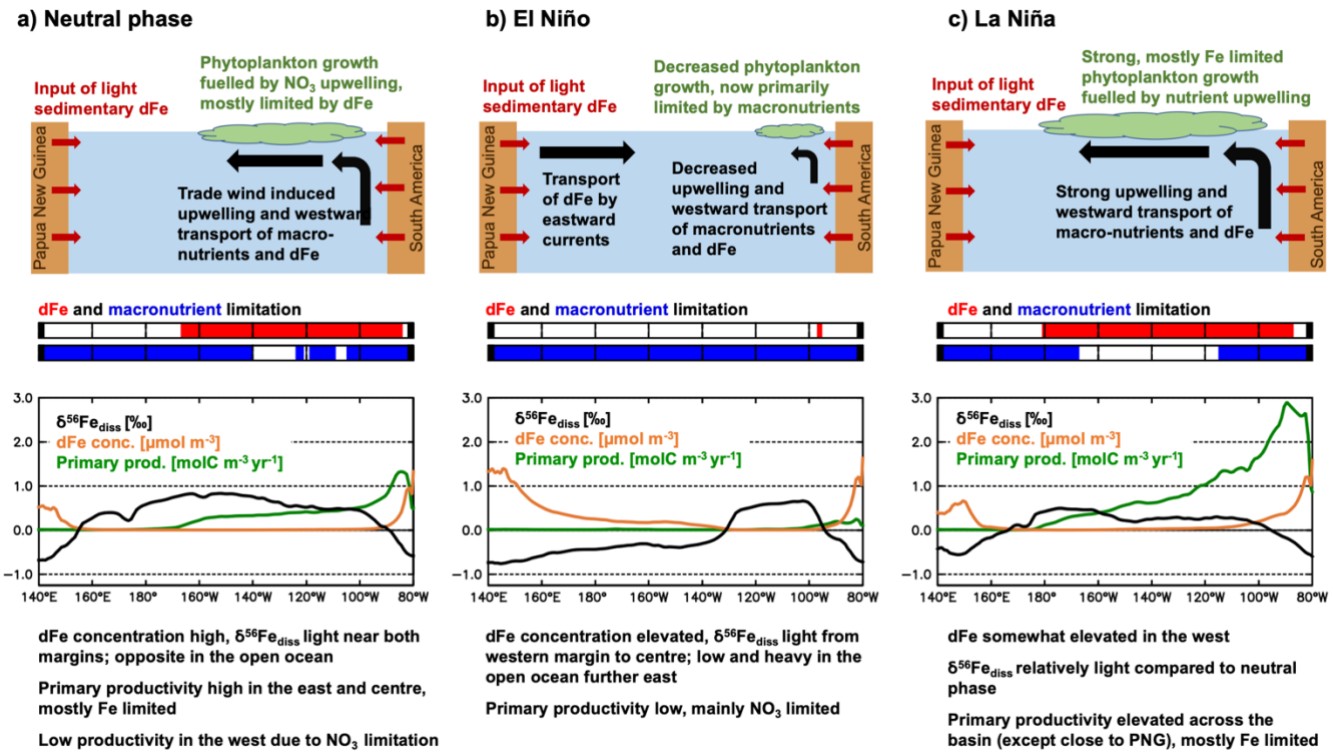

**Figure 4: Overview of processes behind $\delta^{56}Fe_{diss}$ changes in the equatorial Pacific during an El Niño / La Niña cycle. Note that the example $\delta^{56}Fe_{diss}$ (‰), dFe concentration (µmol m$^{-3}$), and primary productivity (molC m$^{-3}$ yr$^{-1}$) curves were taken from the same months as in Fig. 5, i.e., July 1996 for neutral, December 1997 for El Niño, and December 1998 for La Niña phases, using average values from 5°S to 5°N. The dFe and macronutrient limitation plots were taken from the same months and indicate if dFe (red) or macronutrients are limiting phytoplankton growth in (parts of) the 5°S to 5°N region.**

Splitting $\delta^{56}Fe_{diss}$ into its main drivers in the tropical Pacific, $\delta^{56}Fe_{EM}$ and $\delta^{56}Fe_{UF}$ (Fig. 1), shows that the $\delta^{56}Fe_{diss}$ variability during ENSO cycles is due to a combination of endmember and uptake fractionation effects (Fig. 3d,e). These $\delta^{56}Fe_{EM}$ and $\delta^{56}Fe_{UF}$ changes are due circulation changes which affect the redistribution of Fe from external sources and lead to changes in primary productivity and, consequently, Fe uptake rates. We illustrate this with a schematic of the main mechanisms involved (Fig. 4), and use the time period 1995-1996 as an example, as it includes a very strong El Niño followed by a strong La Niña (Fig. 5). During El Niño events, such as in 1997, the weakened trade winds lead to a reversal in the upper ocean currents from predominantly westward flow to eastward flow in most equatorial regions (Fig. 4a,b; Fig. 5h). This reversal has far-reaching effects on upper ocean biogeochemistry and Fe cycling, such as higher dFe concentrations in the west, but lower concentrations in the east (Fig. 5d), and a general decrease in the Fe uptake to dFe concentration ratio (Fig. 5e), Fe limitation (Fig. 5f), and primary productivity (Fig. 5g). These changes, in turn, impact $\delta^{56}Fe_{EM}$ (Fig. 5b) and $\delta^{56}Fe_{UF}$ (Fig. 5c), as discussed below. Eventually, as the El Niño event breaks down, the incoming La Niña leads to westward currents across the basin (Fig. 4c),





which are even stronger than in the neutral phase that precedes the El Niño (Fig. 4a), as exemplified by the transition in 1998 (Fig. 5). Note that the same mechanisms also operate during El Niños or La Niñas that are less strong than those of 1997/98, but to a lesser extent, leading to reduced changes in the parameters in question (Fig. S6). Furthermore, similarly decreasing

trends in dFe concentration and primary productivity as simulated by our model for the equatorial Pacific (Fig. 5) have also been observed in the California Current System during the 1997/1998 El Niño, where coastal upwelling was also suppressed (Johnson et al., 1999).

The reversal of upper ocean currents during El Niño/La Niña cycles has a direct impact on $\delta^{56}Fe_{EM}$. In neutral phases before the on-set of El Niño (e.g., July 1996), waters with elevated dFe concentration are upwelled at the South American margin and

transported westward (Fig. 5d,h). The $\delta^{56}Fe$ endmember signature of this dFe is light, as it is mostly sourced from reducing sediments at the South American margin (Fig. 5b). As the currents reverse during an El Niño event such as in 1997, the upwelling and transport at the eastern margin is reduced and instead, high dFe waters with light $\delta^{56}Fe_{EM}$ from margin sediments in the PNG region are preferentially transported eastwards (Fig. 5b,d). Thus, $\delta^{56}Fe_{EM}$ is light in the west, but remains near a crustal signature (i.e., ca. 0.1‰) in the eastern tropical Pacific, where the main source of external dFe is now dust deposition.

Finally, as the El Niño conditions collapse and a La Niña develops (as in 1998), strong westward currents lead to a similar, but larger, light $\delta^{56}Fe_{EM}$ anomaly from the South American margin as in the neutral phase, which, in the case of the strong El Niño/La Niña cycle of 1997/1998 extends across the basin (Fig. 5b). Note that the sedimentary Fe input from the PNG region may not be as isotopically light as suggested by our model, so that the endmember effects in the western part of the basin may be reduced, as discussed in Sect. 4.1.

The impact of circulation changes associated with the El Niño/La Niña cycle on $\delta^{56}Fe_{UF}$, depend on the ratio of Fe uptake to dFe concentrations. Impacts of El Niño/La Niña on uptake fractionation integrate the changes in primary production, limiting nutrients, and, consequently, Fe uptake rates by phytoplankton. Uptake fractionation becomes strongest in our model when Fe uptake rates are high relative to dFe concentrations (Fig. 5e). This leads to heavy $\delta^{56}Fe_{UF}$. When the ratio between Fe uptake and dFe concentration is high, the system is usually Fe limited (Fig. 5f). During the neutral phases such as in summer 1996,

such "high Fe uptake, low dFe concentration" conditions with heavy $\delta^{56}Fe_{UF}$ are prevalent in the central and eastern tropical Pacific (Fig. 5c,e), which are relatively productive and strongly Fe limited (Fig. 5f,g). As the El Niño develops, the decreased upwelling and westward transport of macronutrients (e.g., nitrate) from the South American margin leads to a decrease in productivity and a switch from Fe to nitrogen limitation in the central and eastern Pacific (Fig. 5f,g). When combined with the input of dFe from the western margin, this shift in nutrient limitation leads to low Fe uptake to dFe concentration ratios (Fig.

5e), and therefore lower uptake fractionation effects (i.e., relatively light $\delta^{56}Fe_{UF}$) across the basin. During the following La Niña, dFe concentrations are relatively high in the Fe-limited eastern and central Pacific (Fig. 5d). These elevated dFe concentrations are due to decreased Fe uptake during the nitrogen limited El Niño phase and the increased westward transport of dFe from the South American margin during La Nina. Consequently, the Fe uptake to dFe concentration ratio is relatively low and uptake fractionation effects moderate (Fig. 5c,e), despite high primary productivity fuelled by the increased upwelling

of nitrate and dFe (Fig. 5g).

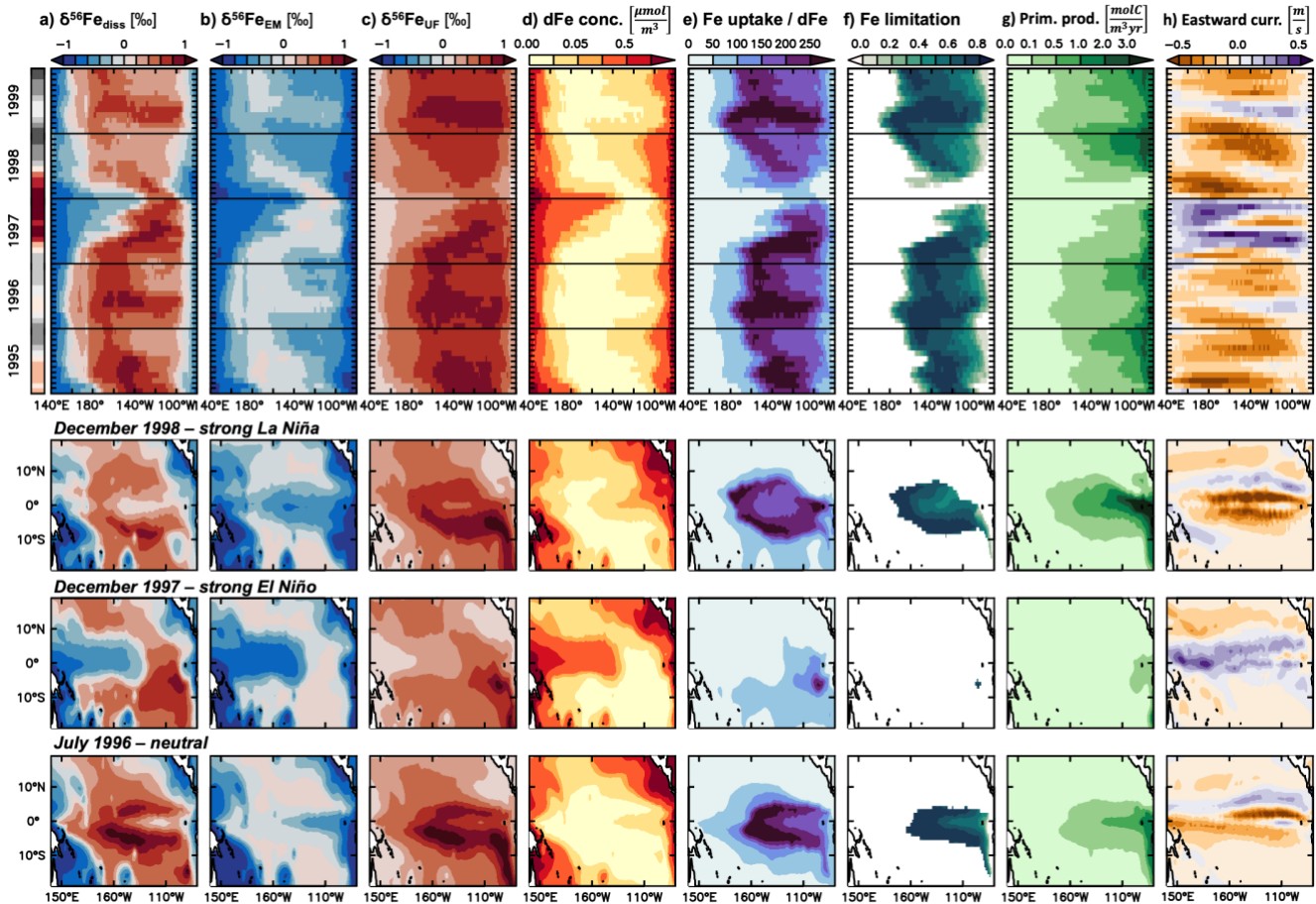

**Figure 5: Mechanisms behind surface ocean δ$^{56}$Fe$_{diss}$ changes in the equatorial Pacific from the hindcast experiment during a strong El Niño / La Niña cycle. Upper half: Time series (1995-1999; from hindcast experiments) of monthly-mean surface ocean (a) δ$^{56}$Fe$_{diss}$ (‰), (b) δ$^{56}$Fe$_{UF}$ (‰), (c) δ$^{56}$Fe$_{UF}$ (‰), (d) dFe concentration (µmol m$^3$), (e) ratio between Fe uptake and dFe concentration, (f) Fe limitation, (g) primary productivity (molC m$^3$ yr$^1$), and (d) upper ocean (0-50m average) eastward currents (m s$^{-1}$), averaged from 5°N to 5°S. The Ocean Nino Index is included on the left side (red: El Niño, Grey: La Niña). Lower half: Map view of the same parameters in July 1996 (neutral), December 1997 (strong El Nino), Dec 1998 (strong La Nina).**

### 3.2.3 Future (2006-2100) δ$^{56}$Fe$_{diss}$ variability in the tropical Pacific driven by climate change

The climate change induced decrease of surface ocean δ$^{56}$Fe$_{diss}$ variability in the western and increase in the eastern equatorial Pacific over the next century shown in Fig. 2 is driven by decreased upwelling of cold water in the east (Fig. S7; Fig. 6, Fig. 7). This reduces the input and westward transport of both macronutrients and isotopically light sedimentary dFe from the South American margin to the surface ocean and decreases primary production. These changes are all similar to those occurring during El Niño events from our hindcast simulations. Consequently, the western half of the equatorial Pacific becomes nitrogen limited and primary productivity low, leading to roughly constant uptake fractionation, i.e., invariable δ$^{56}$Fe$_{UF}$. Conversely, the effects of different ENSO phases are now concentrated in the eastern equatorial Pacific, where variability is high for both





$\delta^{56}Fe_{EM}$ and $\delta^{56}Fe_{UF}$, due to changes in upwelling of light dFe and variability in the degree of Fe limitation (or switches to nitrogen limitation), respectively.

Importantly, circulation patterns during the historical period (1852-2005) in the climate change simulations are not as realistic as for the hindcast experiments, which were forced by an atmospheric reanalysis. For instance, the cold SST anomaly at the

equator is too strong and extends too far west in the climate change simulations, which is a common bias in such climate models (e.g., Planton et al., 2021). This "cold tongue bias" is caused by upwelling in the eastern Pacific which is too strong in the historical era of climate change experiment. This is also evident when comparing equatorial SST of the climate change and hindcast simulation and is likely also responsible for the lower $\delta^{56}Fe_{EM}$ variability in the western tropical Pacific in the climate change simulation (Fig. S7). In addition, there is some disagreement between observations and the ENSO cycles produced by

the climate change model (in the historical period of 1982-2005), for instance, regarding seasonal timing and zonal pattern (Bellenger et al., 2014; Planton et al., 2021). As models which better agree with historical ENSO characteristics predict the frequency of extreme ENSO events to increase (Cai et al., 2015, 2021), future variability in $\delta^{56}Fe_{diss}$ may be higher than predicted by our climate change experiment.

While the physical model shows some biases in terms of SST and ENSO characteristics, the modelled primary productivity

changes in the tropical Pacific in response to changing SST during ENSO cycles agrees well with available constraints from SST observations and satellite estimates of primary production (Kwiatkowski et al., 2017; Tagliabue et al., 2020). This strengthens our confidence in the effect of ENSO cycles on $\delta^{56}Fe_{UF}$, as it indicates that the impact of the different ENSO phases on the model biogeochemistry is realistic. It should be noted that the physical setting of the climate change experiments is based on model simulations using the high emission RCP8.5 scenario. As smaller changes in biogeochemical and physical

conditions are predicted for lower emission scenarios (Dufresne et al., 2013; Kwiatkowski et al., 2020), we would expect the $\delta^{56}Fe_{diss}$ changes also to be reduced in parallel.

### 3.3 Impact of climate change on surface ocean $\delta^{56}Fe_{diss}$ in the southern hemisphere

Future climate change induced alterations to $\delta^{56}Fe_{diss}$ become largest in the southern hemisphere. The largest changes occur a region between about 30 to 50°S (Fig. 2, Fig. 6a) where stronger uptake fractionation effects (i.e., heavier $\delta^{56}Fe_{UF}$; Fig. 6b)

lead to increasingly heavy $\delta^{56}Fe_{diss}$ in the Pacific and Indian ocean (Fig. 6a). These changes to uptake fractionation are ultimately caused by higher primary production rates (Fig. 7c), which lead to stronger Fe limitation (Fig. 7b) and thus higher $\delta^{56}Fe_{UF}$ (Fig. 6b). The increases in $\delta^{56}Fe_{UF}$ between 30 to 50°S are reinforced by an increase in $\delta^{56}Fe_{EM}$ in the western parts of each basin (Fig. 6c) due to a southward shift of low dFe areas with relatively heavy $\delta^{56}Fe_{EM}$ (Fig. 6c, Fig. 7a). Conversely, the $\delta^{56}Fe_{diss}$ decrease at lower latitudes is due to lower primary productivity and Fe limitation (Fig. 7b,c), and thus lighter $\delta^{56}Fe_{UF}$

(Fig. 6b). A notable decrease in $\delta^{56}Fe_{diss}$ thereby occurs just north of the heavy $\delta^{56}Fe_{diss}$ area, due to a decrease in Fe limitation as the region of high, Fe-limited, primary productivity moves southward. Similarly, the strong decrease in $\delta^{56}Fe_{UF}$ (and thus $\delta^{56}Fe_{diss}$) in the South Atlantic is due to a decrease in Fe limitation, and, in some parts, a shift to macronutrient limitation. Many of the changes in dFe and $NO_3$ concentration, primary productivity, and Fe limitation (Fig. 6) that drive $\delta^{56}Fe_{diss}$ changes





over the next century were also observed in the only other available study on the impact of climate change on the ocean Fe

cycle (Misumi et al., 2014). For instance, those authors also report increased future primary productivity at around 40°S, which is Fe limited in the Indian and Pacific, but not anymore in the Atlantic (Misumi et al., 2014). The slight difference in depth range and time periods used in Fig. 6 in this study, relative to those reported by Misumi et al. (2014), has little impact on the trends of each parameter. One major difference between the two simulations concerns primary productivity in the eastern tropical Pacific, which is predicted to decrease by our model, due to a switch from Fe to $NO_3$ limitation, but increases in the

simulation of Misumi et al. (2014), in which phytoplankton remain mostly Fe limited (see also Sect. 4.2 and Tagliabue et al., 2020).

The $\delta^{56}Fe_{diss}$ decrease in some higher latitude areas (around ca. 60°S) is driven by higher concentration of dFe, likely due to redistribution of "new" dFe from external sources. This is consistent with the slight decrease in $\delta^{56}Fe_{EM}$ and the lower $\delta^{56}Fe_{CF}$, which is generally characteristic of "younger" water (Fig. 6c,d; König et al., 2021). Such increases in dFe concentrations also

lower the $\delta^{56}Fe_{UF}$ effect (Fig. 6b), as the Fe uptake to dFe concentration ratio is decreased.

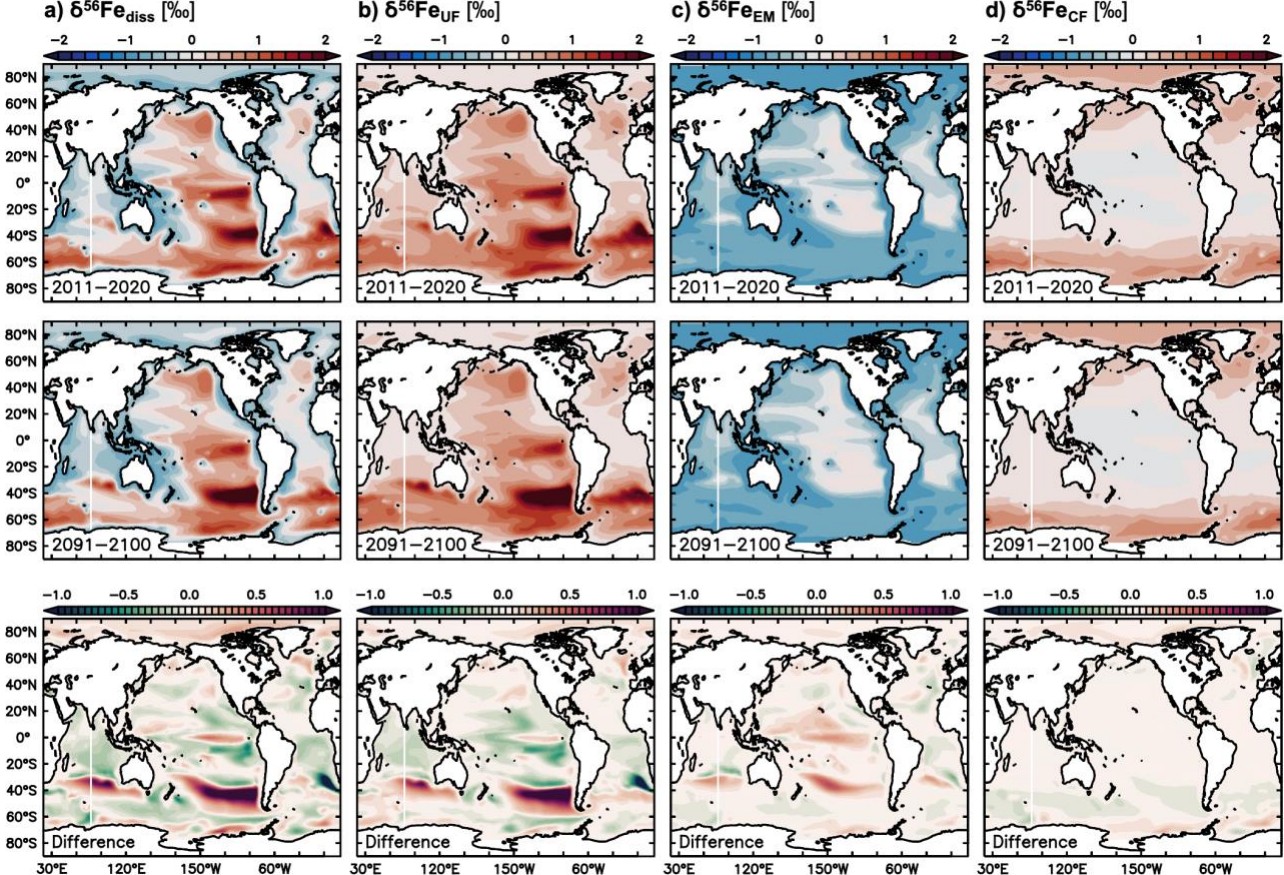

**Figure 6: Climate change effect on $\delta^{56}Fe_{diss}$ and its drivers. Present decade (2011-2020 average) and end of century (2091-2100 average) climate change simulation values of (a) $\delta^{56}Fe_{diss}$ (‰), (b) $\delta^{56}Fe_{UF}$ (‰), (c) $\delta^{56}Fe_{EM}$ (‰), and (d) $\delta^{56}Fe_{UF}$ (‰). The difference of future minus present values is shown in the bottom row.**





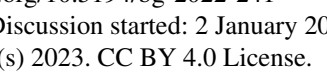

Figure 7: Climate change effect on underlying mechanisms of $\delta^{56}Fe_{diss}$ drivers. Present decade (2011-2020 average) and end of century (2091-2100 average) climate change simulation values of (a) dFe concentration (µmol m³), (b) Fe limitation, (c) primary production (molC m³ yr⁻¹)and (d) nitrate (mmol m³). The difference of future minus present values is shown in the bottom row.

## 4 Wider implications

As discussed above, Fe isotopes have been used in the past to track both external Fe sources and internal cycling. Here, we demonstrated how both historical and future surface ocean $\delta^{56}Fe_{diss}$ variability arises from a combination of the redistribution of "new" Fe from external sources and changes to internal cycling, especially Fe uptake by phytoplankton under Fe limiting conditions. This raises the question as to whether observed variations in $\delta^{56}Fe_{diss}$ can be used to infer alterations in external Fe distribution and Fe uptake and limitation.



### 4.1 Using $\delta^{56}Fe_{diss}$ to track changes in input of new Fe from external sources

Variability in surface ocean $\delta^{56}Fe_{diss}$ in the tropical Pacific is a great example of how $\delta^{56}Fe_{diss}$ may track new Fe input from external sources. Here, $\delta^{56}Fe_{EM}$ are responsible for a substantial fraction of $\delta^{56}Fe_{diss}$ variability (Fig. 1f, Fig. 2d), and track ENSO-related circulation changes that redistribute Fe zonally from different external sources (Sect. 3.2). Specifically, the eastward transport of isotopically light Fe from PNG sediments during El Niño events and the westward transport of light Fe

from Peru sediments are discernible as light $\delta^{56}Fe_{EM}$ (and $\delta^{56}Fe_{diss}$) anomalies in the respective regions (Fig. 3c,d). Simultaneous east and westward shifts in the open ocean region that receives most dust Fe input are also evident based on their more crustal $\delta^{56}Fe_{EM}$ (Fig. 3d).

Tracking external Fe input using $\delta^{56}Fe_{diss}$ is sensitive to the choice of $\delta^{56}Fe$ endmembers for the different (potential) new Fe sources. This has been discussed previously in a range of studies (e.g., Conway and John, 2014; König et al., 2022; Pinedo-

González et al., 2020). For instance, in the tropical Pacific our model includes major new Fe inputs from continental margins, with this sedimentary-sourced Fe being isotopically light, with a $\delta^{56}Fe$ endmember of -1‰ in the upper ca. 400m (as it is assumed to be released by reductive dissolution at these depths independent of location, König et al., 2021). Input of such light $\delta^{56}Fe_{diss}$ appears to be realistic for the eastern margin, as light $\delta^{56}Fe_{diss}$ have been observed in the Peru upwelling region (Chever et al., 2015; Fitzsimmons et al., 2016; John et al., 2018). However, isotopically heavier dFe has been observed in the western

and central tropical Pacific and is suggested to be released from PNG sediments via non-reductive processes with crustal (ca. 0.1‰) $\delta^{56}Fe$ endmember (Labatut et al., 2014; Radic et al., 2011). A sensitivity test with neutral (0‰) sediment $\delta^{56}Fe$ endmember shows without any light sedimentary Fe input, $\delta^{56}Fe_{EM}$ variability and, consequently, $\delta^{56}Fe_{diss}$ variability are reduced in the surface ocean (Fig. S8). This has the consequence of increasing the dominance of fractionation effects (mainly $\delta^{56}Fe_{UF}$; Fig. S8). Thus, in areas such as the western tropical Pacific and areas further east which receive Fe from the Equatorial

Undercurrent (Kaupp et al., 2011), $\delta^{56}Fe_{EM}$ variability may be reduced, as the modelled sediment Fe input from PNG margins is too light. A greater extent of non-reductive Fe input would mean that changes in new Fe input or redistribution by ocean circulation would be harder to detect using $\delta^{56}Fe_{diss}$. This is especially true in areas where fractionation effects are strong and thereby conceal changes in $\delta^{56}Fe_{EM}$. On the other hand, $\delta^{56}Fe_{diss}$ could be useful to track new Fe input and redistribution in areas such as the eastern tropical Pacific, where the $\delta^{56}Fe$ endmember of new Fe is observed to be light, and where $\delta^{56}Fe_{EM}$ is

responsible for a substantial part of $\delta^{56}Fe_{diss}$ $\delta^{56}Fe_{diss}$ variability.

Another complicating factor for tracking Fe input changes with $\delta^{56}Fe_{diss}$ are when fractionation and endmember effects overlap on each other. Most fractionating processes lead to heavier $\delta^{56}Fe_{diss}$ in our model (except for colloidal pumping; König et al., 2021), whereas endmember effects are comparably light, even for dust Fe or Fe from non-reductive sediments with crustal $\delta^{56}Fe$. Thus, if fractionation effects ($\delta^{56}Fe_{UF}$ and/or $\delta^{56}Fe_{CF}$) do not vary in unison with $\delta^{56}Fe_{EM}$, changes in Fe input may be

hidden (as illustrated for the North Pacific in Fig. S2). However, where variability in fractionation and endmember effects occur simultaneously, fractionation effects may reinforce $\delta^{56}Fe_{EM}$ variability. This is mostly the case in the tropical Pacific,





especially at the equator (Fig. S6), with some discrepancies in the regions to the north and south (Fig. S2). Here, changes in the redistribution of external Fe could even be deducted from $\delta^{56}Fe_{UF}$ alone.

In summary, it is necessary to have a good grasp on the relevant $\delta^{56}Fe$ endmembers, when inferring changes in external Fe
input or redistribution from $\delta^{56}Fe_{diss}$, and it is also important to have an idea of how variable $\delta^{56}Fe_{UF}$ and $\delta^{56}Fe_{CF}$ are, compared to $\delta^{56}Fe_{EM}$ (Fig. S1). Such changes are therefore easiest to spot where the external Fe source in question is known to provide isotopically light Fe, and where fractionation effects are weak and/or invariable in time.

## 4.2 Using $\delta^{56}Fe_{diss}$ to track changes in Fe limitation and recycled Fe

Areas of strong Fe limitation are generally associated with heavy $\delta^{56}Fe_{diss}$ driven by strong uptake fractionation effects (i.e.,
heavy $\delta^{56}Fe_{UF}$), often reinforced by relatively heavy $\delta^{56}Fe_{EM}$ due to limited input of light Fe from external sources (Sect. 3.2, Sect. 3.3). This suggests that heavy $\delta^{56}Fe_{diss}$ could be a potentially useful indicator of Fe limitation, which opens up the possibility to detect or even monitor changes in Fe limitation using $\delta^{56}Fe_{diss}$. Such information would be valuable, since Fe is thought to limit primary production in large parts of the global ocean (Moore et al., 2013). However, the specific extent of Fe limitation is unknown and may vary on seasonal, interannual, and decadal timescales. Poorly constrained climate-change
driven changes in Fe limitation have been demonstrated to explain large parts of the uncertainty in model projections of net primary production (NPP) in the tropical Pacific, whereby future NPP is predicted to depend decisively on whether phytoplankton switch from Fe to $NO_3$ limitation, or not (Tagliabue et al., 2020; see also Sect. 3.3).

Outputs from our hindcast experiments show that the connection between heavy surface ocean $\delta^{56}Fe_{diss}$ and Fe limitation generally holds, and that seasonal changes in Fe limitation are tracked by $\delta^{56}Fe_{UF}$ and, consequently, $\delta^{56}Fe_{diss}$ (Fig. S9). As
discussed for the tropical Pacific (Sect. 3.2), this is due to the fact that uptake fractionation is strongest (i.e., $\delta^{56}Fe_{UF}$ heaviest) where the ratio between Fe uptake and dFe concentration is highest, whereby this ratio also determines the degree of Fe limitation (where Fe is the limiting nutrient; Fig. 3, Fig. S5). However, in some regions, namely in the south and south-eastern Pacific, $\delta^{56}Fe_{UF}$ (and $\delta^{56}Fe_{diss}$) can be heavy despite a relatively moderate Fe uptake to dFe concentration ratio (Fig. S9). While we cannot rule out that the heavy $\delta^{56}Fe_{UF}$ is caused by model artefacts, it may also be related to the ocean circulation pattern
of these areas. Whereas in the equatorial Pacific, ENSO dynamics supply water with light $\delta^{56}Fe_{diss}$ every few years, much of the water in southeast Pacific originates in the (seasonally) Fe limited Southern Ocean and therefore contains Fe with relatively heavy $\delta^{56}Fe_{diss}$ (Fig. S9). Here, the heavy $\delta^{56}Fe_{diss}$ may therefore not only be caused by local processes, but could instead by a "legacy" signature. It may also indicate that this dFe has been continuously recycled on its way to the southeast Pacific, which would leave dFe isotopically heavy, with the uptake fractionation parametrisation of our model, since parts of the produced
(isotopically lighter) phytoplankton Fe are eventually removed from the surface ocean via particle settling. When deriving the local degree of Fe limitation from $\delta^{56}Fe_{diss}$ it is therefore important to consider the "background" $\delta^{56}Fe_{diss}$ based on the origin of dFe and previous processing. Moreover, local $\delta^{56}Fe_{EM}$ effects should also be considered, especially if they are variable in time and space.



As discussed above, the connection between heavy $\delta^{56}Fe_{diss}$ and Fe limitation is a consequence of the uptake fractionation

parametrisation of our model. For this parametrisation we use a constant fractionation factor ($\alpha = 0.9995$) for Fe uptake by both phytoplankton classes, based on observations from an Fe-limited, Southern Ocean eddy (Ellwood et al., 2020). Thus, the uptake fractionation strength is independent of factors such as dFe concentration, temperature, or species composition. This is in contrast to uptake of other nutrients such as ammonium, for which fractionation was found to decrease when concentrations were low (Sigman and Fripiat, 2019). While it is conceivable that $\delta^{56}Fe$ fractionation during phytoplankton uptake is impacted

by dFe availability and/or other parameters, observational studies on the topic are limited. Exceedingly heavy surface ocean $\delta^{56}Fe_{diss}$ have been observed in multiple, likely Fe limited systems (e.g., in the eastern tropical Pacific, Southern Ocean, south Atlantic; Fig. S9) and for some of these systems, the heavy $\delta^{56}Fe_{diss}$ has indeed been attributed to continuous biological processing (i.e., uptake and recycling), combined with fractionation during organic complexation (Ellwood et al., 2020; Sieber et al., 2021). This supports the concentration-independent uptake fractionation of our model since heavy surface ocean $\delta^{56}Fe_{diss}$

would not emerge in model simulations if uptake fractionation was set to decrease for low dFe. Nevertheless, more observational or, ideally, experimental data is necessary to determine which factors may impact uptake fractionation and to find the appropriate fractionation factor parametrisations.

Overall, heavy $\delta^{56}Fe_{diss}$ could a useful indicator of Fe limitation if complicating factors such as past processing and variable $\delta^{56}Fe_{EM}$ are taken into account. Fe limitation derived from $\delta^{56}Fe_{diss}$ observations could thereby complement other measures of

nutrient limitation, such as the (scarce) limitation data from incubation experiments (Moore et al., 2013) or the limitation inferred from nutrient deficiencies (Moore, 2016). Our modelling results suggest that changes in Fe limitation can induce strong seasonal variability in $\delta^{56}Fe_{diss}$ in some locations (this study, König et al., 2022), but also variability on interannual and decadal scales. Thus, it would be worthwhile to observe $\delta^{56}Fe_{diss}$ changes over seasonal and interannual time scales in the form of a $\delta^{56}Fe_{diss}$ time series, ideally in a place with variable (degrees of) Fe limitation. A possible candidate location could be the

Southern Ocean Time Series (140°E, 47°S) which is Fe limited in summer (Boyd et al., 2001; Sedwick et al., 1999). Here, our model predicts substantial seasonal variability in surface ocean $\delta^{56}Fe_{diss}$ (between ca. 0.1 to 0.7‰), whereby available observations are within this range (Barrett et al., 2021; Ellwood et al., 2020).

## 5 Conclusion

Simulations of a global ocean model with active $\delta^{56}Fe$ cycle and variable climate forcings show that surface ocean $\delta^{56}Fe_{diss}$

responds distinctly to the changes in ocean physics and biogeochemistry triggered by natural climate variability and long-term global warming effects. Changes in $\delta^{56}Fe_{diss}$ thereby integrate both variations in external Fe supply and redistribution and shifts in upper ocean biogeochemistry (especially presence and degree of Fe limitation), which alter $\delta^{56}Fe$ endmember and uptake fractionation effects, respectively. We therefore suggest that regular observations of surface ocean $\delta^{56}Fe_{diss}$ as part of a long term time series could be useful to track climate-driven changes to upper ocean Fe supply or the degree of Fe limitation in

phytoplankton. As endmember and fractionation effects can overlap, $\delta^{56}Fe_{diss}$ and dFe concentration data should ideally be



accompanied by ancillary measurements to constrain changes in upper ocean circulation or primary productivity. In areas with high seasonal variability in $\delta^{56}Fe_{diss}$ (e.g., due to strong seasonality in primary productivity), a sub-annual sampling interval should be considered.

## Data availability

Model outputs are available from https://doi.org/10.5281/zenodo.7418726

## Author contributions

DK and AT designed the study. DK adapted the model set-up, ran simulations, analysed model outputs, and drafted the manuscript with support from AT. All authors contributed to manuscript editing and revision.

## Competing interests

The authors declare that they have no conflict of interest.

## Acknowledgements

This work was undertaken on Barkla, part of the High Performance Computing facilities at the University of Liverpool, UK. We acknowledge the GTMBA Project Office of NOAA/PMEL for providing the TAO/TRITON moored buoy array dataset. DK and AT received funding from the European Research Council (ERC) under the European Union's Horizon 2020 research 450 and innovation programme (Grant agreement No. 724289).

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
