# Peer review of "The Fingerprint of Climate Variability on the Surface Ocean Cycling of Iron and its Isotopes"

_Biogeosciences, 2022_

## Author Comment (AC1)

This paper discusses the use of iron isotope ratios as a signal of climate change, based on simulations using a model of ocean biogeochemical cycles that includes iron isotope. The authors point out the possibility that changes in the iron cycle associated with ENSO and the global warming can be detected as changes in iron isotope ratios and that iron isotope ratios can be utilized as an indicator of iron limitation, and recommend that regular time series observations be conducted. In addition, the paper discusses possibilities of not being able to detect iron isotope signals when positive and negative anomalies are offset or when isotope ratios in the endmembers are not deviate significantly in the background values. The text is concise and clear, and the analysis is carefully presented. Overall, the paper provides useful insights; thus I recommend to accept this manuscript after minor revision. There are a few points that I have questions about, which are described below. It would be desirable to answer these points and revise the text if necessary.

In the equatorial Pacific, as the authors mensioned, the equatorial undercurrent (EUC), which is a sub-surface flow, transports iron from west to east (e.g. Slemons et al., 2010, GBC), and this iron is thought to be supplied to the surface layer as well. Does this model reproduce well the transport of iron by EUC? Although analysis of this study is limited to the surface layer down to 10 m, the iron cycle in the surface layer is also controlled by the distribution of iron in the sub-surface layer and may be particularly important in the tropical Pacific. Describing how the model can reproduce iron transport by EUC strengthens the reliability of the model results.

We agree that upwelling of iron from sub-surface layers is also important for the distribution of surface ocean iron because the iron transported by the EUC is an additional source to the surface. The average dFe distribution around the equator in our hindcast simulation is similar to the "control" simulation presented by Slemons et al. (2009), which does not include additional Fe sources in the western Pacific but agrees better with biogeochemical data (chl, NO3) in the eastern equatorial Pacific. Moreover, the elevated dFe within the EUC 'core' along 140°W of around ~1nM in the Slemons paper simulations that included an enhanced deep source to the EUC appear to be strongly overestimated, relative to other data. To illustrate this, and to compare the hindcast simulation to the available data in this region, we plotted dFe data from Slemons et al. (2009, 2010) and other studies (Coale et al., 1996; Fitzwater et al., 1996; Kondo et al., 2012; Mackey et al., 2002; Wu et al., 2010, Kaupp et al. 2011) on top of model outputs along the equator (0-300m) and at 140°W (2°S to 10°N), similar to Figures 3 and 4 in Slemons et al. (2009):

[Figure]

dFe concentration at equator · dFe concentration at 140°W

a) Average (1975-2021) [nM]
d) Average (1975-2021) [nM]
b) Maximum (1975-2021; monthly resolution) [nM]
e) Maximum (1975-2021; monthly resolution) [nM]
c) Minimum (1975-2021; monthly resolution) [nM]
f) Minimum (1975-2021; monthly resolution) [nM]

◆ Slemons, et al. 2010   ■ Coale, et al. 1996   ● Mackey, et al. 2002   ▼ Kaupp, et al. 2011.   ⬠ Kondo, et al. 2012   ⬡ Wu, et al. 2010
+ Takeda & Obata 1995   ▲ Fitzwater, et al. 1996

While our model does a good job in reproducing data along 140°W, we may underestimate dFe concentrations within the EUC in the western tropical Pacific (although we note there is a lot of variability across studies). We will add a comment in the discussion (Section 4.1) to introduce this caveat and include this figure in the supplement:

"Also, the sedimentary Fe supply from the PNG region to the equatorial undercurrent (EUC) may be underestimated in our model compared to some of the available regional observations(but not all, Figure SX). As the EUC supplies Fe to the surface ocean (e.g., Coale, et al 1996, Kaupp, et al. 2011), this may cause additional impacts on  surface ocean dFe concentrations and their response to climate variability (Fig. 5d), as well as on the response in $\delta^{56}Fe_{EM}$ and, consequently, $\delta^{56}Fe_{diss}$."

I understand that the positive iron isotope anomaly around 40S in the South Pacific (Fig. 1b) is due to strong uptake fractionation effects (Fig. S1d). I, however, do not understand why the uptake fractionation effects are stronger in this region. Adding supplementary explanations will help readers' understanding.

The explanation of this feature is discussed around line 395 in the original manuscript:

”..the water in southeast Pacific originates in the (seasonally) Fe limited Southern Ocean and therefore contains Fe with relatively heavy $\delta^{56}Fe_{diss}$ (Fig. S9). Here, the heavy $\delta^{56}Fe_{diss}$ may therefore not only be caused by local processes but could instead be a "legacy" signature. It may also indicate that this dFe has been continuously recycled on its way to the southeast Pacific, which would leave dFe isotopically heavy, with the uptake fractionation parametrisation of our model, since parts of the produced (isotopically lighter) phytoplankton Fe are eventually removed from the surface ocean via particle settling."

We agree that this should be mentioned/referred to in the discussion of Figure 1 and are planning to revise the manuscript as follows (line 157 in original manuscript):

"The elevated $\delta^{56}Fe_{diss}$ SD at around 40°S is caused by horizontal shifts of areas with heavy $\delta^{56}Fe_{diss}$ caused by strong (accumulated) uptake fractionation effects (Fig. 1c, Fig. S1d), as discussed in Section 4.2.

For the hindcast simulation, there are no figures regarding the basic quantities (e.g., distribution of primary production, dissolved iron concentration and the isotope ratios, etc.). The standard deviation of the interannual variation of iron isotope ratios is suddenly shown as the first figure. I think that showing the figures for the basic quantities first will help the readers' understanding of the model results and authors' arguments. It should be noted that the most readers have no prior knowledge about the characterisity of this model.

Figure 1b included the average iron isotopic signature for the hindcast simulation, average values for iron limitation and iron uptake as a fraction of dFe are included in Figure S9. However, we did not include figures of all modelled tracers and fluxes. To respond to the reviewers suggestion, we will add another supplementary figure presenting average and standard deviations for dissolved Fe and primary production, which will be referred to in section 3.1.1:

[Figure]

If regular observations of iron isotope ratio were possible, the obtained data would be time series data at a certain station. The authors could strengthen their arguments if they select one station from the model results, as an example, and show how the use of iron isotopic ratio helps to explain interannual variations of dissolved iron concentration.

The goal of this paper was not necessarily to design new time-series, but instead to examine the interannual variability in iron and iron isotope cycling. The comment about time series observations in this paper at the end of the 'wider implications' section (lines 418 onwards) was to indicate the potential usefulness of regular Fe isotope measurements in areas of (intermittent) Fe limitation, especially on seasonal scales. We chose not to include time series plot at any particular stations, since *interannual* variability at traditional time series stations (e.g., SOTS, BATS, HOT, PAPA) is relatively low compared to areas such as the equatorial Pacific presented in Figures 3-5:

[Figure]

Instead, we have decided to amend this text to refer back to the model results explicitly, so that readers can consider other time series locations, including sites for potential new studies.

New text (l. 425 of original manuscript):

"Here, our model predicts substantial seasonal variability in surface ocean $\delta^{56}Fe_{diss}$ (between ca. 0.1 to 0.7‰, Figure 1a), whereby available observations are within this range (Barrett et al., 2021; Ellwood et al., 2020). Moreover, based on predicted present and future variability in Fe limitation and, consequently, $\delta^{56}Fe_{diss}$ in our model (Fig. 1-2), the equatorial and south eastern Pacific could be potential locations for future studies that explored the interannual changes in iron limitation."

The following are really minor comments.

In Table 1, a case of "Hindcast neuSED" is mentioned, but the result is not presented as a figure in the main text. Since the description of this case in the text (L. 356) is enoughly understandable regarding the model setting, there is no need to list it as a case in Table 1. In addition, in the "d56Fe endmembers" column of the "Hindcast neuSED" case, there is description of "Sediments: -1 per mill,", which may be a mistake of "0 per mill".

Will be updated.

I think the abbreviation ONI in line 202 stands for Ocean Nino Index, so it should be written as (dark red, Ocean Nino Index; ONI).

Will be updated.

In line 221, there is a description of "d56FeEM and d56FeUF (Fig. 1)". I think there is no need to refer "Fig. 1" here because d56FeEM and d56FeUF are clearly defined in Equations (1) and (3).

Will be updated.

Observations of iron isotopic ratios are currently very restricted, especially with little known about interannual variability, and it will be a long time before the findings of this study are validated. Seasonal variations in dissolved iron have been shown in several studies, I felt that it would have been easier and more useful to use this isotope model to evaluate the factors that cause seasonal variations of dissolved iron concentrations. I am looking forward to seeing such a study in futre.

We agree with the reviewer that studying the drivers of seasonal variability in dFe concentration and signatures would be worthwhile, but unfortunately beyond the scope of this study. We are planning work in this area in the future.

---

## Author Comment (AC2)

The authors applied a global ocean model describing the Fe and 56Fe cycling to study the impact of climate variability on surface distribution of iron concentration and its isotopic signals. The model considers different isotopic compositions of sources and fractionation by biological uptake and organic complexation of iron. Their previous publication (König et al. 2021) presented the modelled distribution of delta56Fe and a thorough comparison with observations. In this study this model was driven by different climate forcing data and a series of sensitivity experiments were conducted to quantify the contribution of single factors to the inter annual variability of delta56Fe. Strong responses of delta56Fe to climate change were found in the model. I find the idea to study climate variability with Fe isotope fingerprints highly interesting and the article was well-written and easy to follow. However, I have some concerns about the analysis of model results and kindly ask the authors to give explanations for the following points:

1. Line 120-123: The effect of two single components, fractionation by biological uptake and organic complexation, on delta56Fe is estimated from the difference between an experiment with all components switched on and another one with only one component switched off (Eq. (1) and (2)). But the effect of the third component, isotopic compositions of endmembers, is estimated in a different way (Eq. (3)) which assumes that the three components act independently on delta56Fe in a linear relationship which is not true. An experiment with all endmembers set to 0 ‰ is to my opinion necessary to disentangle the effect of all single components, as the authors mentioned themselves as well (L. 125-126). If this experiment was already done I would like to see if the result is identical to the estimation presented now in the manuscript and why.

We did not originally perform additional experiments with endmembers set to 0‰, since the way our model is set up, the impacts of uptake/complexation fractionation and endmember effects add up linearly. To confirm this, we have now run such a simulation (for the hindcast set-up) and compared the corresponding $\delta^{56}Fe_{EM}$ (calculated similarly as in eq. 1,2) to the "residual" $\delta^{56}Fe_{EM}$, as described in eq. 3. This comparison shows that, beyond rounding errors, the calculated $\delta^{56}Fe_{EM}$ and SD $\delta^{56}Fe_{EM}$ are the same (within 0.002‰ and 0.0005‰, respectively) - confirming the linearity and our original approach.

[Figure]

a) Interannual SD δ⁵⁶Fe_EM 1975-2021 (0-10m) [‰]
δ⁵⁶Fe_EM calculated as outlined in eq. 3

b) δ⁵⁶Fe_EM calc. based on separate simulation [‰]

c) Difference: a) minus b) [‰]

d) Average δ⁵⁶Fe_EM 1975-2021 (0-10m) [‰]
δ⁵⁶Fe_EM calculated as outlined in eq. 3

e) δ⁵⁶Fe_EM calc. based on separate simulation [‰]

f) Difference: d) minus e) [‰]

We agree that the linearity of the endmember and fractionation effects is not obvious, and will add this figure to the supplement and reference the figure in the main text (line 122 of the original manuscript):

"Thanks to the additive nature of fractionation and endmember effects in our model, which we confirmed for the hindcast experiments (Figure SX), the endmember effect $\delta^{56}Fe_{EM}$ could be calculated by subtracting the two fractionation effects (Eq. 3) from $\delta^{56}Fe_{diss}$."

2. Line 135-138: If I understand it correctly, the authors calculated SD of each distribution of delta56Fe resulted from Eq. (1) to Eq.(3) and then the fraction of each single SD in the sum of them. SD can demonstrate the variability around the mean state but tells nothing about the mean state itself. Responses of the three single components to the interannual climate variability can be reflected in SD but also in the mean state of delta56Fe. So I don't quite understand why just SD of different runs are used to examine the contribution of single components.

We agree with the reviewer that climate variability causes substantial changes in the mean state of $\delta^{56}Fe_{diss}$ and its components, especially over the longer time scales of the climate change simulations. However, since the standard deviation was calculated over the entire

time period (i.e., not relative to a running interannual mean), changes in the mean state are accounted for, and are, indeed, responsible for the majority of "variability" in the climate change simulations (e.g., compare Fig. 2a vs. Fig. 6 in the submitted paper). We do agree that the contribution of mean state changes to temporal variability over the 21$^{st}$ century should also be emphasised, and will include this in Section 3.1.2 (l. 169):

"Whereas over the shorter period of the hindcast experiments (1975-2021), elevated $\delta^{56}Fe_{diss}$ SD is mainly due to temporal variability around a mean $\delta^{56}Fe_{diss}$ value, for the climate change experiments, elevated $\delta^{56}Fe_{diss}$ SD is also related to a change in the mean $\delta^{56}Fe_{diss}$ over the next century (Fig. 6a)."

Furthermore, the sum of three SDs is not the same as SD delta56Fediss of the experiment with all components switched on, due to the non-linear relationship between the single components and different signs of the effects. The authors only discussed about the latter in the manuscript. I have no doubt that the results of the three experiments are interesting and can help us to understand how the marine Fe isotope cycle responses to climate variability. Different SDs of the three distributions indicate that each component is differently sensitive to climate variability. However, the interpretation of the relative importance in percentage needs a justification.

The earlier response hopefully persuades the reviewer that the system is linear, and we discussed how overlaps and variability on different frequencies contributed to the results in the original manuscript. The percentage plot is aimed to illustrate what effect can be considered to be dominant in different ocean areas.

At this stage I would like to encourage the authors to revise the analysis and interpretation of the model results. After that, I would be happy to provide more detailed comments.

We have responded to all the comments provided by the reviewer (as well as those proposed by reviewer 1) and hope that this has provided the necessary reassurance to the reviewer.

---

## Author Response (AR2)

**Response to review**

In my last comment there were two questions to be explained: the linearity of the system and the usage of standard deviation in the analysis of climate variability.

For the first question the authors included an additional experiment with the endmember effect set to 0 and used it to confirm the linearity. I agree that the system seems to be linear and the differences between two calculation ways are negligible. However, I expected an explanation why a system with processes which obviously interact work approximately in a linear way. I suppose that not the model setup causes the linearity but the nature of the Fe isotope system. A good example for this approximation is the system of carbon isotopes (e.g. Hayes 1982). Therefore, I suggest the authors to make it clear if and which model setup leads to the linearity or for which system such an approximation can be made.

Hayes, J.M., 1982. Fractionation et al.: An introduction to isotopic measurement and terminology. Spectra, 8(4), 3–8.

We are glad that the reviewer agrees that we have demonstrated that the system is linear. The references to the Hayes summary was not clear, but seems to ask whether, as for carbon, we can sum describe the isotope system as a linear sum of different components or whether we have imposed this linearity through our model parameterisations.

Ultimately, the set up of the model is responsible for this linearity in the modelled iron isotope system and we have observed this for all different experimental set-ups tested so far. In responding to this, we note that fractionation factors are constant (i.e., independent of Fe concentration) and only applied to Fe transferred between two Fe pools for each time step, i.e., there is no equilibration between Fe isotope pools. Similarly, source endmembers are constant (in time) and applied to the input of Fe independent of flux strength or other factors. To exemplify this approach, a simplified version of how light ($^{54}Fe_{diss}$) and heavy ($^{56}Fe_{diss}$) tracers of dissolved Fe are modelled as shown below (eq. 1-4) was added to the supplement. Note that Fe supply by other external sources are modelled similarly to dust supply, and other non-fractionating internal transformation processes are modelled similarly to remineralisation. Fractionation during Fe complexation is modelled similarly as uptake fractionation whereby the inverse of its fractionation factor is applied to the scavenged Fe flux, as scavenging only removes uncomplexed Fe. Other tracers (phytoplankton Fe, Fe particles) are modelled similarly as dissolved Fe, although without any external supply.

$$\frac{\partial\,^{56}Fe_{diss}}{\partial t} = \underbrace{J_{dust,Fe} * \frac{R_{dust,Fe}}{R_{dust,Fe}+1}}_{\text{External Fe supply}} + \underbrace{J_{remin,Fe} * \frac{R_{Fe,part}}{R_{Fe,part}+1}}_{\text{Int. trans. without fractionation}} - \underbrace{J_{up,Fe} * \frac{R_{Fe,diss} * \alpha_{up}}{R_{Fe,diss} * \alpha_{up}+1}}_{\text{Int. transf. with fractionation}} \quad (1)$$

$$\frac{\partial\,^{54}Fe_{diss}}{\partial t} = J_{dust,Fe} * \frac{1}{R_{dust,Fe}+1} + J_{remin,Fe} * \frac{1}{R_{Fe,part}+1} - J_{up,Fe} * \frac{1}{R_{Fe,diss} * \alpha_{up}+1} \quad (2)$$

$$R_{dust,Fe} = \frac{^{56}Fe_{IRMM-014}}{^{54}Fe_{IRMM-014}} * \left(\frac{\delta^{56}Fe_{dust}}{1000} + 1\right) \quad (3)$$

$$R_{Fe,part} = \frac{^{56}Fe_{part}}{^{54}Fe_{part}} \quad (4)$$

$$R_{Fe,diss} = \frac{^{56}Fe_{diss}}{^{54}Fe_{diss}} \quad (5)$$

| | |
|---|---|
| $J_{dust,Fe}$ | dissolved Fe supply rate from dust |
| $R_{dust,Fe}$ | Isotope ratio of supplied dust Fe |
| $\delta^{56}Fe_{dust}$ | Endmember signature of dust Fe supply |
| $J_{remin,Fe}$ | dissolved Fe supply by remineralisation of Fe particles |
| $R_{Fe,part}$ | Isotope ratio of Fe particles, see eq. 5 |
| $^{56}Fe_{part}$ | Heavy particulate Fe concentration |
| $^{54}Fe_{part}$ | Light particulate Fe concentration |
| $J_{up,Fe}$ | dissolved Fe uptake by phytoplankton |
| $R_{Fe,diss}$ | Isotope ratio dissolved Fe, see eq. 4 |
| $\alpha_{up}$ | fractionation factor for phytoplankton Fe uptake |
| $^{56}Fe_{diss}$ | Heavy particulate Fe concentration |
| $^{54}Fe_{diss}$ | Light particulate Fe concentration |

Note that the linearity of the iron isotope system may be challenged by developments in the modelling of iron isotopes in the future and new work (based on either refined modelling or new observations) may depart from these assumptions. Unfortunately, a discussion of this linearity, and where or when it may not apply, is beyond the scope of this study, which instead focusses on interannual variability in the cycling of iron, iron limitation and iron isotopes. But to respond to the reviewers comment, we have added the full system of equations and associated text to the supplement.

Unfortunately the second question can not be answered with the linearity. By combining independent variables (the central limit theorem), standard deviations of variables can not be added together but variances. Therefore, Eq. 4 is wrong. Here I suggest to use the root of the sum of variances of three distributions instead of the sum of SDs, and redo the analysis of climate variability.

We agree with the reviewer that the calculation of the "contribution" of $\delta^{56}Fe_{EM}$, $\delta^{56}Fe_{UF}$, and $\delta^{56}Fe_{CF}$ to $\delta^{56}Fe_{diss}$ as outlined in equation 4 is not ideal. However, we note that using the root of the sum of the three variances would also lead to misleading values, since there is a degree of covariance between the three components, as their temporal response to climate variability can be aligned (or opposing). Thus, we decided to use the ratio between

the variance of each component to the variance of $\delta^{56}Fe_{diss}$ as an indicator of their relative importance in contributing to $\delta^{56}Fe_{diss}$ variability, and also show the covariance between $\delta^{56}Fe_{EM}$ and $\delta^{56}Fe_{UF}$, as they are generally more variable than $\delta^{56}Fe_{CF}$. Note that we did not replace standard deviation with variance for plots that illustrate the variability in $\delta^{56}Fe_{diss}$, dFe concentration and primary productivity (Fig. 1a, Fig. 2a,b, Fig. S2) as it allows for easier comparison between the variability of a parameter and its average, or between the degree of variability for two model experiments.

We updated Section 2.3 as follows:

Line 137 of the revised manuscript:

"To compare the interannual variability of each of the three components to $\delta^{56}Fe_{diss}$ variability, we calculated their "relative variability", i.e., the ratio between their VAR and that of $\delta^{56}Fe_{diss}$, where subscript i=UF, CF or EM (Eq. 4).

$$\delta^{56}Fe_i \text{ relative variability} = \frac{VAR(\delta^{56}Fe_i)}{VAR(\delta^{56}Fe_{diss})} \hspace{3cm} (4)$$

Note that the sum of the three ratios are lower (higher) than 1 in areas with positive (negative) covariance between two or three of the components, i.e., where their response to climate variability are reinforcing (opposing) each other."

We also updated the sections 3.1 and 4 of the manuscript accordingly (see tracked version of the manuscript), namely by adding the relative variability of each component and the covariance between $\delta^{56}Fe_{EM}$ and $\delta^{56}Fe_{UF}$ to Figures 1, 2, S5, and S11. Note that this new approach left Fig. S3 and Fig. S6 (which showed standard deviations of drivers) obsolete, so that they were removed. We also updated Fig. S4 (now Fig. S3) to show both examples of negative and positive covariance between parameters. Note that the major results and conclusions drawn are not changed.